# Index analysis: An approach to understand signal transduction with application to the EGFR signalling pathway

**Jane Knöchel**[1,2¤], **Charlotte Kloft**[3], **Wilhelm Huisinga**[1]\*

**1** Institute of Mathematics, Universität Potsdam, Potsdam, Germany, **2** Graduate Research Training Program PharMetrX: Pharmacometrics & Computational Disease Modeling, Freie Universität Berlin and Universität Potsdam, Berlin/Potsdam, Germany, **3** Department of Clinical Pharmacy and Biochemistry, Institute of Pharmacy, Freie Universität Berlin, Berlin, Germany

¤ Current address: AstraZeneca R&D, Mölndal, Sweden
\* huisinga@uni-potsdam.de

**Data Availability Statement:** All data are in the manuscript and/or supporting files. The MATLAB codes used for index analysis are available at https://zenodo.org/record/8393545.

## Abstract

In systems biology and pharmacology, large-scale kinetic models are used to study the dynamic response of a system to a specific input or stimulus. While in many applications, a deeper understanding of the input-response behaviour is highly desirable, it is often hindered by the large number of molecular species and the complexity of the interactions. An approach that identifies key molecular species for a given input-response relationship and characterises dynamic properties of states is therefore highly desirable. We introduce the concept of index analysis; it is based on different time- and state-dependent quantities (indices) to identify important dynamic characteristics of molecular species. All indices are defined for a specific pair of input and response variables as well as for a specific magnitude of the input. In application to a large-scale kinetic model of the EGFR signalling cascade, we identified different phases of signal transduction, the peculiar role of Phosphatase3 during signal activation and Ras recycling during signal onset. In addition, we discuss the challenges and pitfalls of interpreting the relevance of molecular species based on knock-out simulation studies, and provide an alternative view on conflicting results on the importance of parallel EGFR downstream pathways. Beyond the applications in model interpretation, index analysis is envisioned to be a valuable tool in model reduction.

## Author summary

In systems biology, the response of signalling networks to various molecular stimuli is studied through large and complex kinetic models. Prominent examples include the epidermal growth factor receptor signalling cascade and its response to growth factors or therapeutic interventions by small molecules and therapeutic protein drugs. The design of targeted interventions requires a detailed understanding of the signalling cascade, the identification of key molecular constituents and their functional role in propagating the signal. A quantitative analysis of such systems, however, is challenging due the size of the

**Funding:** J.K. kindly acknowledges financial support from the Graduate Research Training Program PharMetrX: Pharmacometrics & Computational Disease Modelling, Berlin/Potsdam, Germany. The funders had no role in study design, data collection and analysis, decision to publish, or preparation of the manuscript.

**Competing interests:** The authors have declared that no competing interests exist.

model and the complexity of molecular interactions. We present an approach based on time- and state dependent quantities that are called indices and quantify different characteristic dynamic features of the molecular constituents. This allows for a more in-depth understanding of how the signal propagates through the network and elicits its response. Our findings for the epidermal growth factor receptor signalling cascade provide new insights on the dynamic interplay of key molecular players.

## Introduction

Large scale models of biochemical reaction networks are increasingly used to study the dynamical response of a system to a specific input. The response is often measured in terms of some output quantity of interest. Examples include the therapeutic inhibition of the epidermal growth factor receptor (EGFR) signalling cascade in cancer [1, 2] or the development of anti-coagulant drugs targeting the blood coagulation network [3]. Understanding the input-response relationship for a given model, however, is a challenging task when the order (dimension) of the model is large and the interactions between its constituents are manifold. We introduce a novel approach to characterise the dynamic property of state variables within signalling networks by means of time- and state dependent quantities, called indices. By design, all indices are specific for a given input-response relationship and a given magnitude of the input.

The sensitivity-based input-response index allows to quantify the dynamical importance of states. It is based on the product of two local sensitivity coefficients; the first coefficient quantifies the impact of the input on a given state variable at a given time, while the second coefficient quantifies how a perturbation of a given state variable at a given time impacts the output on the remaining time interval. States having a large input-response index are considered dynamically important. To further characterise states with a low input-response index, we define the environmental and the partial steady state classification indices. They are based on a comparison of the original kinetic system to a modified kinetic system and allow to identify very slow state variables (environmental) or very fast state variables (partial steady state). Two additional state classification indices allow to study the impact of removing/neglecting state variables in the network.

Approaches to study complex systems biology models include sensitivity and flux analysis [4], model reduction techniques [5–10] and model interpretation approaches [11–14].

Sensitivity analysis quantifies how a change in parameters or initial states influences the system output, and several different variants have been developed [4]. Based on sensitivity coefficients, parameters or initial states are classified as important or unimportant for a given model output [15], but in particular appropriate scaling of the coefficients for comparability remains a challenge, see also Discussion. To account for the inherently transient behaviour of signalling transduction, time-dependent sensitivity coefficients have been considered, including more robust sensitivity coefficients based on local averaging [14]. Often, however, these are subsequently converted to time-independent quantities by integration in time, facilitating comparison and accounting for delays in signal propagation that often render pointwise evaluations meaningless [13]. An alternative approach is pursued in [12] by probabilistically identifying dominant reaction paths combining topical geometry and ultradiscretization theory.

For model reduction techniques, a common interpretation is to classify state variables that are part of the reduced model as important and those that are not as unimportant. Different model reduction techniques such as time-scale separation or balanced truncation exploit

different underlying characteristics of the system. In time scale separation, the system dynamics needs to exhibit different time scales, where a partitioning into slow and fast states and a subsequent quasi-steady state approximation of the fast variables results in the reduced model [16]. In balanced truncation [17], the basic idea is to transform the system into its principal components and neglect the least important components while still maintaining the same input-response behaviour as the original system. Application to signalling networks, however, are limited [5]. Profile likelihood based model reduction utilises parameter identifiability to designate likely parameter candidates for reduction [10]. It is a powerful tool for model reduction, focussing on experimental data and parameters, rather than on states variables and time. There appear to be interesting links between the different scenarios of parameter non-identifiability and our proposed indices.

Model interpretation approaches additionally add to our understanding of complex system biology models. Topological sensitivity analysis extends classical sensitivity analysis from the impact of parameter variations to the impact of changes in model architecture [11], which has implications on the biophysical interpretation of model parameters. In [14], the dependence of parameter sensitivity on the chosen response and condition (input) has been studied shedding light on the interpretation of robustness of parameters. To dissect the relative contribution of different reaction channels with overlapping molecular species to a common readout (response), causal tracing of signal flows based on in silico molecular labelling is used in [13] to interpret the observed paradoxical activity in drug-inhibited BRAF mutant melanoma in terms of rewiring of the signalling cascade.

Despite this repertoire of approaches to study complex systems biology model, our means remain limited to study the *time-resolved* relevance of reactions and molecular species during signal. As concluded in [14], there is a need for "the development of suitable descriptors of time-varying transfer properties in cell signalling cascades". An attempt to study the time-resolved relevance of parameters ("At what time is a given parameter relevant?") is presented in [18], but many questions remain. Limited approaches exist to study the time-resolved relevance of molecular species. Typically, these rely on some a-priori assumed decomposition of the signalling network into modules and a focus on some key molecular species to reduce the complexity [13]. The presented index analysis aims to address this gap by answering the questions: At what time(s) is a molecular species relevant for the transduction of the signal? And why?

In [19], we introduced the concept of an empirical input-response index by building and expanding on concepts from control theory (empirical controllability and observability gramians). Given some input and response, the empirical index quantifies the time-dependent relevance of a state variable for the given input-response relationship based on some finite set of perturbations. We illustrated in a clinically relevant setting, how the empirical index can guide us to reduce a large systems biology model of the blood coagulation network by eliminating states variables, either by completely removing them from the system or by assuming them to be constant. In the present article, we substantially extend the concept of an index and at the same time focus on understanding a given signal transduction network rather than reducing it. The index analysis is first introduced based on a number of simple model systems to ease understanding of the different index types and their application. We finally illustrate the power of index analysis in application to a large-scale model of the EGFR signalling cascade [1, 20]. This signalling network has been intensively studied in the context of tumour genesis and tumour progression as well as for anti-cancer treatment strategies, e.g., to overcome drug resistance in non-small cell lung cancer [21]. From a signal transduction point of view, the EGFR network remains a challenging system due to the large number of interacting molecular species and its multiple parallel pathways.

## Methods

Many dynamical models in systems biology/pharmacology are of the form

$$\frac{\mathrm{d}x}{\mathrm{d}t}(t) = f(x(t); p), \quad x(t_0) = x_0 + u_0 \tag{1}$$

$$y(t) = h(x(t)) \tag{2}$$

with $x(t) \in \mathbb{R}^n$ denoting the vector of state variables at time $t \in [t_0, T]$, and $p \in \mathbb{R}^m$ denoting the vector of parameters. In the sequel we assume that all parameter values are fixed, but the approach can be extended to distributions of parameter values accounting for variability (see Discussion). The function $f : \mathbb{R}^n \times \mathbb{R}^m \to \mathbb{R}^n$ represents the reaction kinetic model, and is typically of the form

$$f(x; p) = \sum_{\mu=1}^{M} v_\mu \alpha_\mu(x; p) \tag{3}$$

with stoichiometric vectors $v_\mu \in \mathbb{R}^n$ and reaction rates $\alpha_\mu \in \mathbb{R}$ of the reactions $\mu = 1, \ldots, M$. The function $h : \mathbb{R}^n \to \mathbb{R}^q$ maps $x(t)$ to the output $y(t)$ of interest, often only a single state variable. The initial condition $x(t_0)$ comprises two parts: (i) the state $x_0 \in \mathbb{R}^n$ of the system prior to the stimulus, and (ii) an (delta) input or stimulus $u_0 \in \mathbb{R}^n$ at time $t_0$. An extension to time-dependent, continuous inputs is possible. The solution of the ordinary differential equations (ODEs) in Eq (1) is written as

$$x(t) = \Phi^{t,t_0}(x_0 + u_0) \tag{4}$$

with state transition function $\Phi^{t,t_0}$. The output then takes the form

$$y(t) = h(\Phi^{t,t_0}(x_0 + u_0)). \tag{5}$$

The assumptions on $f$ (time-independence) and $h$ (independent on parameters $p$) can be relaxed.

In the sequel, we used both, $z_k$ and $[z]_k$ to denote the $k$th entry of a vector $z = (z_1, \ldots, z_n)$; the choice depended on whatever was deemed easier to read. An analogous notation was used for matrices.

### Input-response index

The (sensitivity-based) input-response index quantifies the dynamic importance (to be defined below) of state variables. The input-response index for a state $x_k$ at time $t^* \in [t_0, T]$ is based on two factors that characterise to what extent

(i) perturbations in the input $u_0$ impact the state variable $x_k$ at time $t^*$

(ii) perturbations in the state $x_k$ at time $t^*$ impact the output $y$ on the remaining time interval $[t^*, T]$.

It is well-known that signalling cascades might respond differently to different magnitudes of the same input variable. Consequently, we defined the input-response index relative to some reference input $u_{\mathrm{ref}}$. It defines a reference trajectory

$$x_{\mathrm{ref}}(t) = \Phi^{t,t_0}(x_0 + u_{\mathrm{ref}}); \qquad t \in [t_0, T] \tag{6}$$

according to Eq (1). To quantify the extent to which a perturbation of the input impacts a

given state variable, we considered a perturbed input $u_{\text{per}} = u_{\text{ref}} + \Delta u_{\text{per}}$ and quantified the impact using a first-order Taylor approximation (indicated by the dot on top of the "=" sign)

$$x_{\text{per}}(t^*) = \Phi^{t^*,t_0}(x_0 + u_{\text{per}}) \doteq \Phi^{t^*,t_0}(x_0 + u_{\text{ref}}) + J_u(t^*, t_0)\Delta u_{\text{per}} \tag{7}$$

with Jacobian

$$J_u(t^*, t_0) = \frac{\partial}{\partial u} \Phi^{t^*,t_0}(x_0 + u)\Big|_{u=u_{\text{ref}}} = \frac{\partial x_{\text{ref}}(t^*)}{\partial u}.$$

Relative to the reference input, this resulted in a perturbation

$$\Delta x_k(t^*) = \left[x_{\text{per}}(t^*) - x_{\text{ref}}(t^*)\right]_k \doteq \left[J_u(t^*, t_0)\Delta u_{\text{per}}\right]_k$$

of the $k$th state variable at time $t^* \in [t_0, T]$. The larger $[J_u(t^*, t_0)]_{k,i}$ the stronger the $i$th input impacts the $k$th state variable at time $t^*$. If $[J_u(t^*, t_0)]_{k,i} = 0$, then the $i$th input has no impact on the $k$th state variable at time $t^*$. This quantification alone, however, is not sufficient to infer the importance of state variables for a specific input-response relationship, since it only quantifies the first aspect (i) above.

Thus, we next quantified the extent to which a perturbation of the $k$th state variable at $t^*$ impacts the output $y$ on the remaining time interval. The key idea is to reinterpret the perturbation $\Delta x_k(t^*)$ as an input

$$u_{\Delta x_k}(t^*) = (0, \dots, 0, \Delta x_k(t^*), 0, \dots, 0) \in \mathbb{R}^{k-1} \times \mathbb{R} \times \mathbb{R}^{n-k}$$

to the model system in Eq (1) with (unperturbed) initial condition $x_{\text{ref}}(t^*)$ at initial time $t^*$. This resulted in the perturbed output (to first order Taylor approximation):

$$y_{\text{per},\Delta x_k(t^*)}(t) = h\left(\Phi^{t,t^*}(x_{\text{ref}}(t^*) + u_{\Delta x_k}(t^*))\right) \doteq h\left(\Phi^{t,t^*}(x_{\text{ref}}(t^*))\right) + J_y(t^*, t_0)u_{\Delta x_k}(t^*) \tag{8}$$

with $t \in [t^*, T]$. Realising that

$$y_{\text{ref}}(t) = h\left(\Phi^{t,t_0}(x_0 + u_{\text{ref}})\right) = h\left(\Phi^{t,t^*}(x_{\text{ref}}(t^*))\right), \tag{9}$$

we thus quantified the resulting perturbation on the $j$th output component as

$$\left[y_{\text{per},\Delta x_k(t^*)}(t) - y_{\text{ref}}(t)\right]_j \doteq \left[J_y(t^*, t_0)u_{\Delta x_k}(t^*)\right]_j$$

with Jacobian

$$J_y(t, t^*) = \frac{\partial h(\Phi^{t,t^*}x)}{\partial x}\Big|_{x=x_{\text{ref}}(t^*)}.$$

Analogously as above, the larger $[J_y(t, t^*)]_{j,k}$ the stronger the $k$th state variable at time $t^*$ impacts the $j$th output during the remaining time interval. If $[J_y(t, t^*)]_{j,k} = 0$, then the $k$th state variable has at time $t^*$ no impact on the $j$th output.

We finally defined the sensitivity-based input-response index $\text{ir}_k(t^*)$ of the $k$th state variable at time $t^*$ as

$$\left[\text{ir}_k(t^*)\right]_{ji} = \left(\frac{1}{T}\int_{t^*}^{T}\left([J_y(t, t^*)]_{j,k}\right)^2 dt\right)^{\frac{1}{2}} \cdot [J_u(t^*, t_0)]_{k,i}. \tag{10}$$

The definition is motivated from control theory (see also below). The first factor represents the time-average integrated impact of an (infinitesimal) state perturbation over the time span $[t^*, T]$ on the output; the integral form guarantees that a transient impact during the time span is taken into account, even if it occurs only during some short time span in the interval $[t^*, T]$. If, however, only the output at a single event time $t_{\text{event}}$ is of interest, then $[J_y(t_{\text{event}}, t^*)]_{j,k}$ rather than the integral should be considered. For time-dependent inputs, also the second factor becomes an integral; for delta-type inputs as in our case it reduced to a single value at the initial time.

In general, $\text{ir}_k(t^*)$ is a matrix of dimension (number of outputs) × (number of inputs). For the common situation of a single state input (i.e., $u_0$ has only a single, say $i$th non-zero entry) and a single response state (i.e., $y(t) = h(x(t)) = x_r(t) \in \mathbb{R}$ for some state index $r \neq i$), the input-response index is real-valued and can be written in terms of two local sensitivity coefficients

$$\text{ir}_k(t^*) = \left(\frac{1}{T}\int_{t^*}^{T}\mathcal{S}_{r,k}(t;t^*)^2\mathrm{d}t\right)^{\frac{1}{2}} \cdot |\mathcal{S}_{k,i}(t^*;t_0)| \tag{11}$$

with sensitivity coefficients

$$\mathcal{S}_{m,j}(t_2;t_1) = \left[\mathcal{S}(t_2;t_1)\right]_{m,j} = \left[\frac{\partial \Phi^{t_2,t_1}x}{\partial x}\bigg|_{x=x_{\text{ref}}(t_1)}\right]_{m,j}. \tag{12}$$

It is a distinct feature of the input-response index that it combines both, the impact of the input on a state variable as well as the impact of the state variable on the output. Neither one nor the other on its own is in general informative to quantify the relevance of a state variable for a given input-response relationship. In a control theoretical setting, the factors

$$\mathcal{O}_k(t^*) = \left(\frac{1}{T}\int_{t^*}^{T}\mathcal{S}_{r,k}(t;t^*)^2\mathrm{d}t\right)^{\frac{1}{2}} \qquad \text{and} \qquad \mathcal{C}_k(t^*) = \mathcal{S}_{k,i}(t^*;t_0) \tag{13}$$

can be interpreted as a controllability index $\mathcal{C}_k$ and an observability index $\mathcal{O}_k$.

To ease comparison of the ir-indices for different points in time, we normalised each index by the sum of all indices, resulting in the normalised ir-indices (nir):

$$\text{nir}_k(t^*) = \frac{\text{ir}_k(t^*)}{\text{sum} - \text{ir}(t^*)}; \qquad \text{sum} - \text{ir}(t^*) = \sum_{j=1}^{n}\text{ir}_j(t^*). \tag{14}$$

As a result, the nir-index takes only values between 0 and 1. The sum of ir-indices is also of interest, as it gives some indication on the overall magnitude of the ir-values. Details on the numerical computation can be found in S1 & S2 Supplementary Materials.

We classified a state variable as dynamically important, if its input-response index is above a (user-defined) threshold at some point in time. A threshold of $\delta_y = 10\%$ was successfully used in all examples of the results section. The relevance of the input-response index is two-fold: (i) it allows to characterise the dynamic pattern of importance of a state variable: When is a state variable important for the input-response relationship? At which times is it less important? And (ii) it allows to identify state variables that are not dynamically important and to further characterise them.

## State classification indices

The motivation of additional state classification indices is to further characterise states with a small input-response index. In such a case, the impact of the input on a state variable is

insufficient to see a marked change in the output. This includes the two extreme cases, that the input has no impact on that state variable, or that the state variable has no impact on the output. The former would typically be considered an environmental state variable, the latter be considered negligible. It can be shown that also state variables with very fast dynamics have a low input-response index.

We introduced four state classification indices ($env_k$, $pss_k$, $pneg_k$ and $cneg_k$). Each state classification index measures the impact of a modification (mod) of the original system at time $t^*$ on the output during the remaining time interval $[t^*, T]$. The specific type of modification defines the corresponding state classification index. For the $k$th state variable, the common definition of all four state classification indices and the $j$th output is

$$\left[\text{ind}_k(t^*)\right]_j = \left(\frac{1}{Z_j(t^*)} \int_{t^*}^T \left(\left[y_{\text{ref}}(s) - y_{\text{mod}}^{(k)}(s)\right]_j\right)^2 \mathrm{d}s\right)^{1/2} \tag{15}$$

with $t^* \in [t_0, T]$ and normalisation constant $Z_j(t^*) = \int_{t^*}^T \left|[y_{\text{ref}}(s)]_j\right|^2 \mathrm{d}s$. Here and below, $x_{\text{mod}}^{(k)}(s)$ and $y_{\text{mod}}^{(k)}(s)$ denote the solutions of the modified system of ODEs at time $s \geq t^*$. See also S1 Supplementary Material for an example.

- The *environment index* ($env_k$) quantifies to what extent the $k$th state variable can be classified as an environmental state, defined as being constant in time. Thus, the modification of the system of ODEs at time $t^*$ is to set the right hand side (RHS) of the $k$th ODE to zero.

- The *partial steady state index* ($pss_k$) quantifies to what extent the $k$th state variable can be classified as being instantaneous in steady state; since this only affects the $k$th state variables, it is a partial steady state of the system. The modification of the system of ODEs at time $t^*$ is to set the left hand side of the $k$th ODE to zero, resulting in a system of differential-algebraic equations (DAEs). Thus, the dynamics of the $k$-state variable is characterised by an algebraic equation rather than on ODE. Many numerical solvers can integrate this type of equations. Of note, we used the term 'partial steady state' to differentiate it from a quasi-steady state, which typically involves the exploitation of additional conservation laws. For further illustration and clarification, see the analysis of the enzyme kinetics model given in the Section S4.2 in S4 Supplementary Material.

- The *partially and completely neglected indices* ($pneg_k$, cneg) quantify to what extent the $k$th state variable can be neglected and thus removed from the system. This can be done with two different intentions, as illustrated for the reaction system A ⇌ B ⇀ C. Assume (i) that C is just a degradation product with no further relevance. So we want to remove C from the system while maintaining the important degradation of B, resulting in A ⇌ B ⇀ *. This corresponds to the partially neglect index. In contrast, assume (ii) that the exchange between *A* and *B* is fast, but $A \ll B$. If *A* is of no further relevance, we might want to neglect it. Just removing A without its producing reaction would, however, result in an additional fast degradation of *B*, or * ⟵ B ⇀ C. So, here, we want to remove **A** and all related reactions, resulting in B ⇀ C. This corresponds to the completely neglect index.
  Thus, for (i) the *partially neglected index* $pneg_k$, the modification to the system of ODEs is to set all reaction rate constants to zero that involve the $k$th state variable as *reactant* species. In many reaction kinetic systems, this can easily be realised by setting $x_k(t) = 0$ for all $t \in [t^*, T]$. For (ii) the *completely neglected index* $cneg_k$, the modification to the system of ODEs is to set all reaction rate constants to zero that involve the $k$th state variable *as reactant or product* species.

Due to the type of normalisation, the normalised state classification indices can take values larger than 1. Details on the numerical computation of the four state classification indices can be found in S1 & S2 Supplementary Materials, incl. a pseudocode.

The state classification indices only measure the impact (of a specific change of the system of ODEs) on the output. For a final classification of a state variable, it is, however, important to also measure the impact on the state variable itself. Thus, for the state classification indices $\text{env}_k$ and $\text{qss}_k$, we additionally defined a corresponding relative state approximation error

$$\text{rel} - \text{state} - \text{err}_k(t^*) = \left( \frac{1}{Z(t^*)} \int_{t^*}^{T} \left| [x_{\text{ref}}(s) - x_{\text{mod}}^{(k)}(s)]_k \right|^2 \, ds \right)^{1/2} \tag{16}$$

with $t^* \in [t_0, T]$ and normalisation $Z(t^*) = \int_{t^*}^{T} |[x_{\text{ref}}(s)]_k|^2 \, ds$. Note that for the indices $\text{pneg}_k$ and $\text{cneg}_k$, the relative state approximation error is not meaningful, since it equals 1 by definition.

We classified a state variable as environmental or in partial-steady state, if the corresponding state classification index *and* the state approximation error are both below (user-defined) thresholds for all times $t \in [t_0, T]$. We classified a state variable as partially or completely negligible, if the partially/completely neglected index is below a (user-defined) threshold for all times $t^* \in [t_0, T]$. In all examples, we used the same threshold of $\delta_y = 10\%$ for all state classification indices, and also $\delta_x = 10\%$ for state approximation errors. We note that both threshold can be chosen independently and adapted to any specific needs. Fig 1 summaries the strategy we used to classify states.

The MATLAB code of the index analysis approach with application to the illustrative model systems and the EGFR system is available on zenodo at URL https://zenodo.org/record/8393545. New models can very easily implemented, see S3 Supplementary Material.

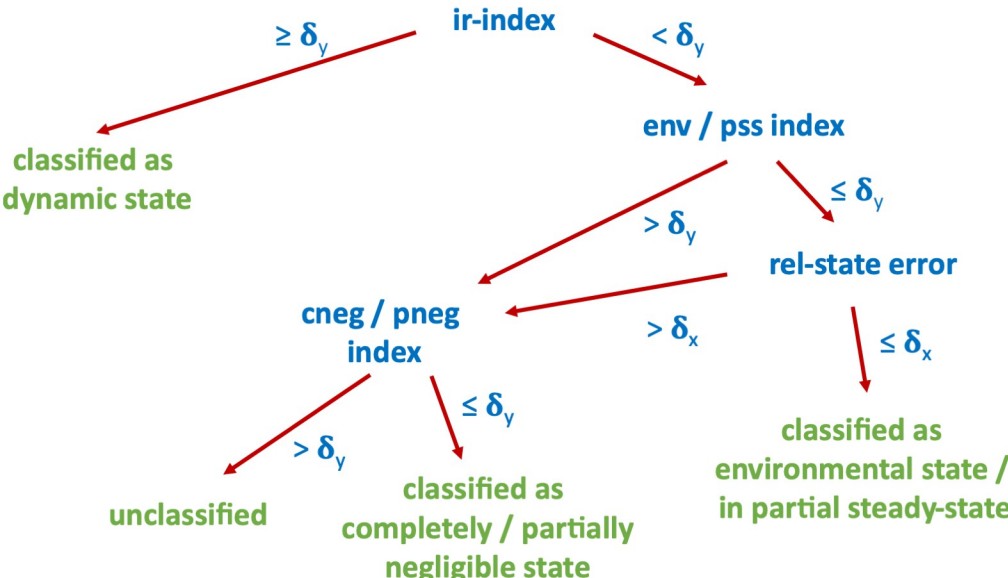

**Fig 1. Decision tree for state classification based on indices.** See text for details.

## Material

We chose a number of simple model systems to illustrate the application, interpretation and usefulness of the indices, of which one is included in the Results section and the remaining are included in the S4 Supplementary Material.

We then studied the epidermal growth factor (EGF) receptor signalling network [1, 20, 22] to illustrate how the indices allow to obtain detailed insights into the dynamic behaviour of complex, large-scale systems biology/pharmacology models. The EGFR system is an an important pathway in cell division, death, motility and adhesion [1, 23, 24]. In addition, it is of key interest in the development of anti-cancer therapies, as the pathway is often dysfunctional in tumour cells. We used a detailed model of the EGFR reaction network [20] consisting of 106 state variables and 148 reactions. We followed [20] regarding the abbreviations of state variable names. The original model includes a lumped pseudo state of degradation products that serves as a substitute for various individual degradation products. To allow for a more refined analysis of receptor degradation, we modified the original model by separating the degraded receptor species (EGF- EGFR*)$_2$- deg into six separate degradation products, increasing the number of state variables by six to 112. This extension does not change the remaining system dynamics. All initial conditions and parameter values for the model were taken from [20]. In addition, all corrections reported in [25] were taken into account.

In contrast to our expectations, the system published in [20] is not in steady state in the absence of EGF, i.e., the stimulus of the system. Some state variables do change, including EGFR, EGFRi, Grb2, Sos and Grb2-Sos; see Fig II in S5 Supplementary Material. To ensure comparison to the original publication in [20], however, we did not make any further changes.

## Results

### Index analysis for an illustrative model system

To illustrate the index analysis in application, we first considered a simple reaction cycle model sketched in Fig 2. In the model, the signal A transforms B into C, which in turn activates D. In addition, A, C and D may be subject to degradation. In this example, A is considered the input and D the response variable. We distinguished three different scenarios defined by the parameter values in the table in Fig 2B. These were chosen to illustrate how the indices allow

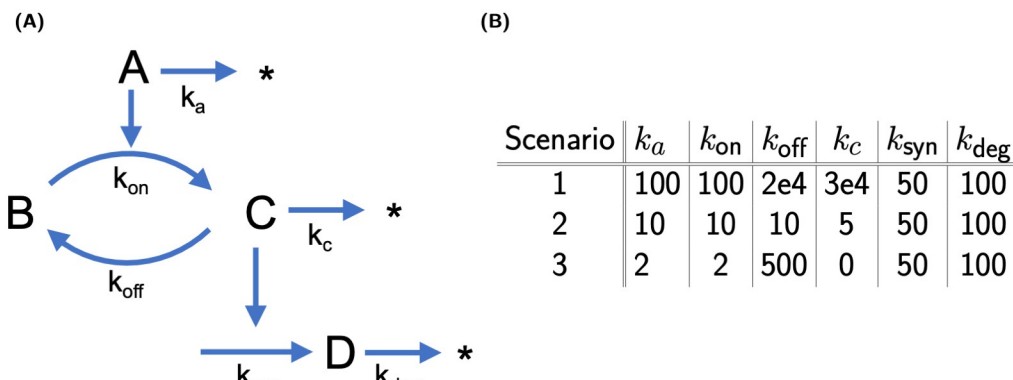

**Fig 2. Simple reaction cycle model and scenario-specific parameter values.** (A) Reaction network, and (B) parameter values for the three scenarios, for which the model system was studied. The time span was $t \in [0, T]$ min with $T = 0.01$ or 0.1 (depending on the scenario), the input variable A, and the response variable D. The initial conditions were identical in all three scenarios: (A, B, C, D)$_0$ = (0, 100, 5, 0) nM and input $u_0$ = (2, 0, 0, 0) nM. Units: $k_{on}$ in 1/nM/min; all other reaction rate constants in 1/min.

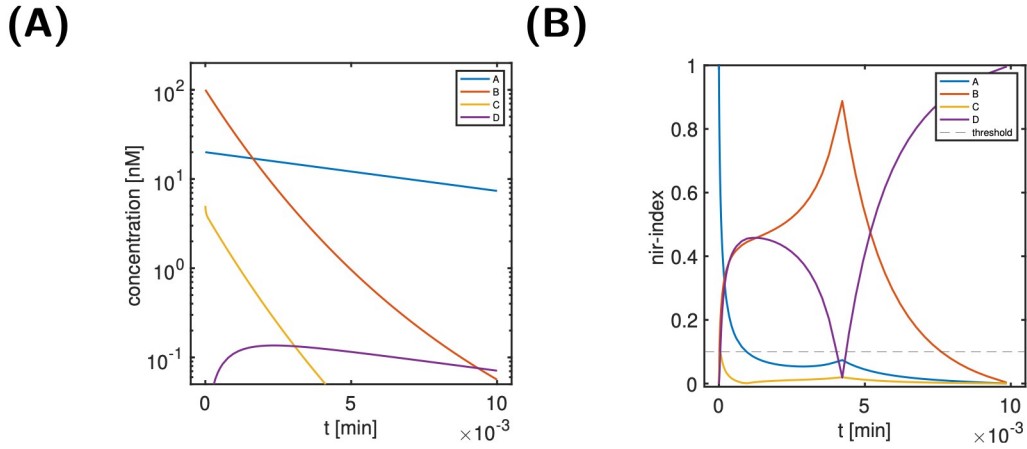

**Fig 3. Scenario 1 of the simple reaction cycle model: Time course of state variables (Panel A), and normalised ir-indices (Panel B).** For details on the model, see Fig 2.

to analyse and differentiate between the scenarios. We use Scenario 1 to introduce the indices in detail and build on this in Scenarios 2 and 3. We used the decision-tree in Fig 1 to guide the analyses.

Fig 3A shows the time course of the state variables, while Fig 3B shows the normalised ir-indices. Here and in all other analyses, we empirically (as with many thresholds) used a threshold of 10%; this choice was supported *a-posteriori* by the results. One nicely observes from panel B that the dynamic importance of state variables changes substantially over time: Initially, the signal *A* has the largest dynamic importance; it decays, however, very quickly and below a threshold of 10%. In contrast, the indices of all other state variables are negligibly small initially and increase steeply. While B and D reach interim values of 45% around 2 min, C increases only up to 10%, before it starts to decays again. It stays below 10% for the entire time span. The dynamic importance of B and D continue to evolve with almost opposite behaviour. Eventually, B decays and D increases to its maximal value. For a classification of states, the particular features of the index are not considered relevant; the key feature considered is whether an index stays below the threshold for all times, or not. For the input-response index, we considered a state dynamically unimportant, if its ir-index stays below the threshold of 10% for the entire time span. Otherwise, we consider it as 'not unimportant', i.e., as dynamically important. Thus, we considered A, B and D as dynamically important, but not C.

The top graphics of Fig 4 show the state classification indices for two states, B (left) and C (right). These indices are designed to further analyse the characteristics of dynamically unimportant states, i.e, here state C. We also included state B, though dynamically important, for illustration only. A state classification index below the threshold for all times is considered indicative of the corresponding index classification type (environmental, in partial steady state, negligible).

For a state classification as environmental or in partial-steady state, also the relative state approximation error (shown in Fig 4) is taken into account. Since the normalised input-response indices for B was above the threshold, we would expect to see no further characteristics for state B. In contrast, given the below-threshold normalised input-response indices for C, we would hope to get further indications for its classification. In line with these expectations, none of the state classification indices for B is below the threshold, while for C a single

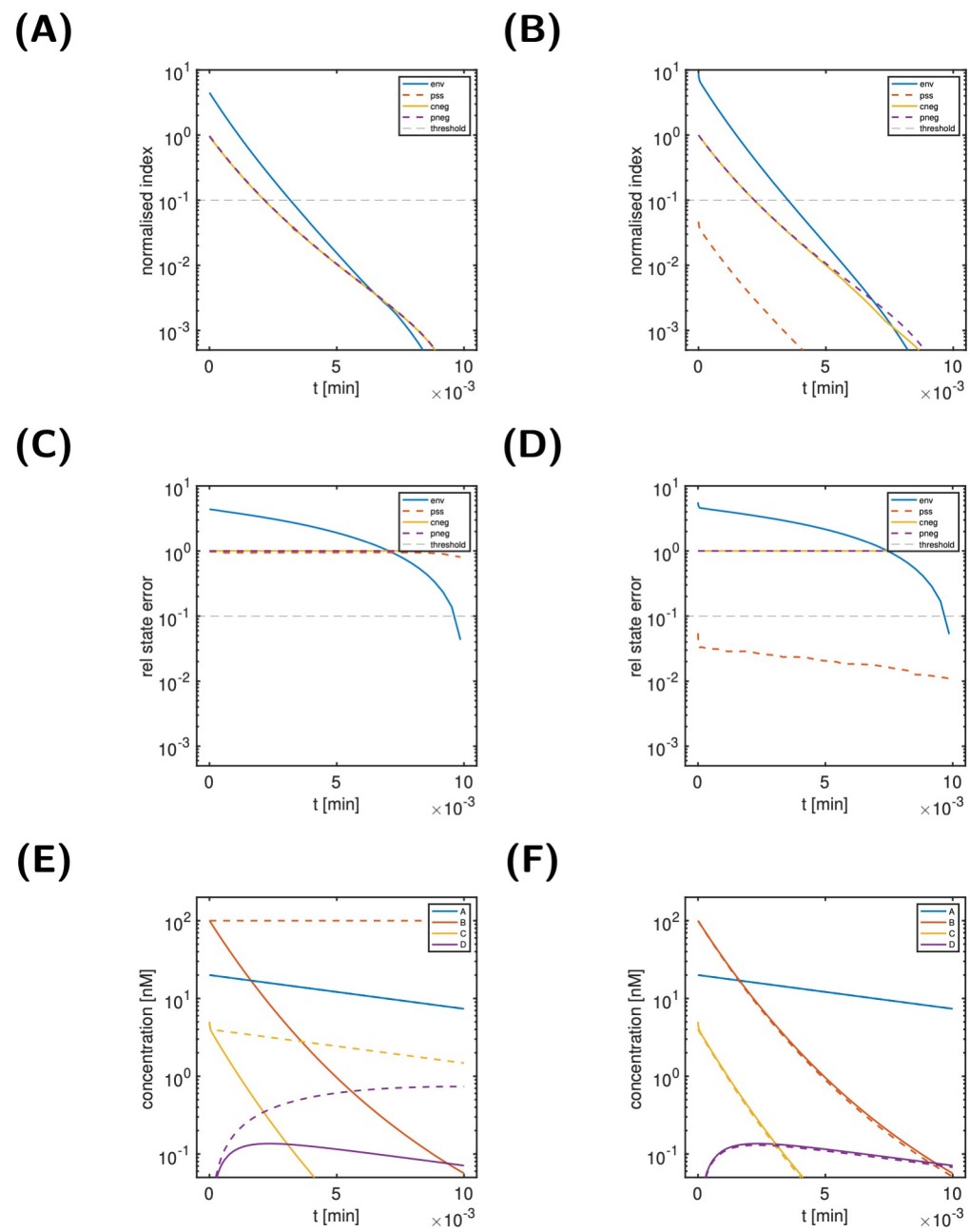

**Fig 4. Scenario 1, states B (left) and C (right) of the simple reaction cycle model: State classification indices (Panels A, B), relative state approximation error (Panels C, D) and modified dynamics (Panels E, F).** Panels E and F: Comparison of reference dynamics and modified dynamics from $t^* = 0$ (dashed lines) for B as environmental state and for C in partial steady state. For details on the model, see Fig 2.

index (the partial steady state index) is below the threshold for all times. This is an indication for C being in a partial steady state. This is finally confirmed by Fig 4D. The middle panels show the relative state approximation errors for state B (left) and C (right). We infer that only the classification of C being in partial steady state (red dashed line) results in an approximation error below the threshold for all times. The bottom-right panel illustrates the impact of

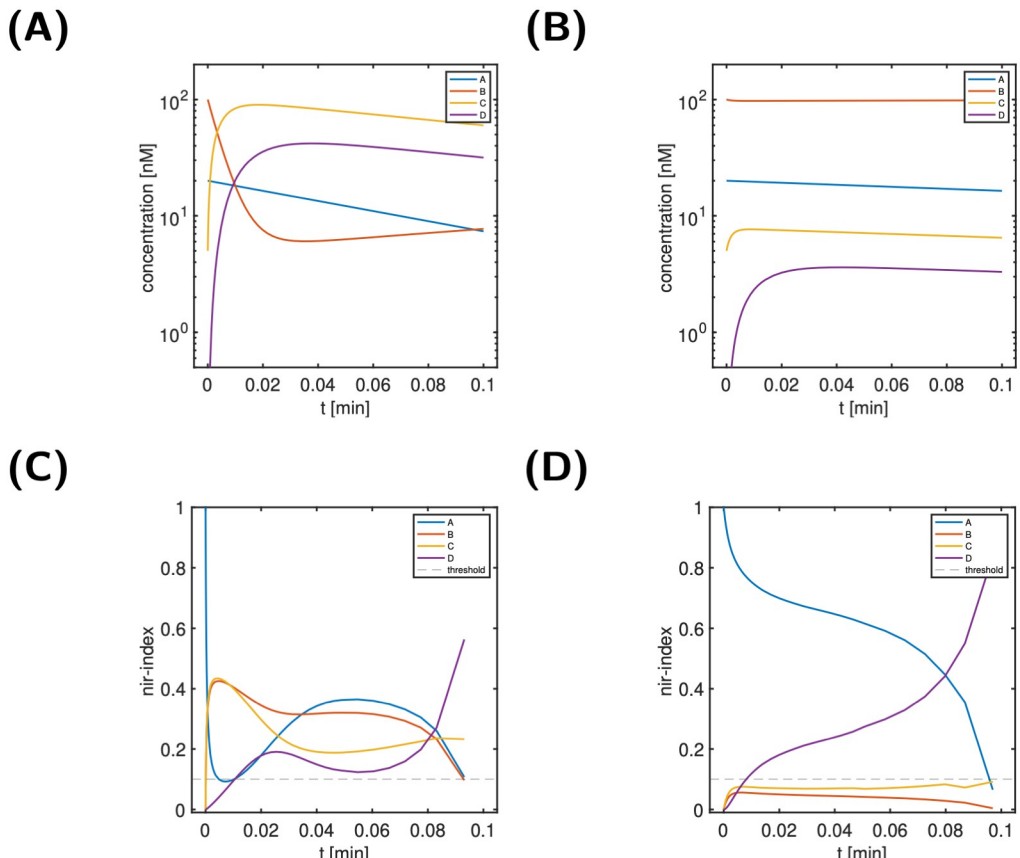

**Fig 5. Scenario 2 (left) & 3 (right) of the simple reaction model: Time course of state variables (Panels A,B), and normalised ir-indices (Panels C,D).** For details on the model, see Fig 2.

modifying the reaction system to enforce C being in partial steady state from time $t^* = 0$ onwards. As can be seen, the solution of the modified system (dashed lines) is a very good approximation to the solution of the reference model (solid lines). For sake of illustration, the bottom left panel show the impact of modifying the system so that B is environmental (i.e. constant) from $t^* = 0$ onwards. Clearly and in line with the large environmental index of B, the resulting approximation is very poor. Thus, we conclude for Scenario 1 that C can be considered in partial steady state, while all other states are considered as dynamically important.

For Scenarios 2+3, Fig 5 shows the time course of all state variables and the normalised input-response indices. For Scenario 2 (left column), we conclude from Fig 5C that all states are classified as dynamically important, since no nir-index is below the threshold for all times. In contrast, for Scenario 3, we conclude from Fig 5D that only states A and D are dynamically important, while states B and C are not. Fig 6 (top) shows the state classification indices for B (left) and C (right). Two indices are below the threshold for all times: the env-index for B and the pss-index for C. The bottom panel finally confirms this classification. The relative state approximation errors for B as environmental state (left, solid blue line) and C as in partial steady state (right, dashed red line) are below the threshold for all times.

For Scenario 2, we conclude that all states are dynamically important. For Scenario 3, in contract, only A and D are dynamically important, while B is an environmental state and C is in partial steady state.

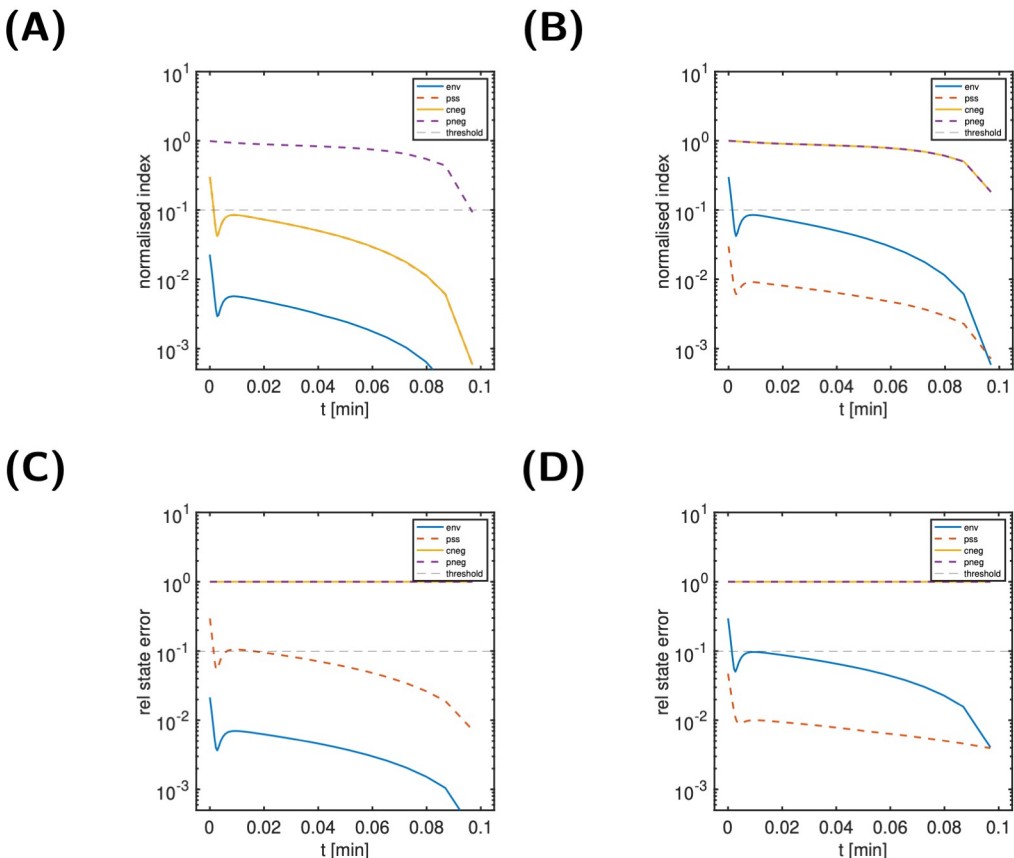

**Fig 6. Scenario 3, states B (left) and C (right) of the simple reaction model: Normalised state classification indices (Panels A, B) and relative state approximation errors (Panels C, D).** For details on the model, see Fig 2.

The input-response index classifies whether or not a state variable is considered dynamic for a particular set of parameters and given input-output relationship. The four state classification indices provide further understanding in which way the state variable with a low input-response index (below 10%) impact the output either as a constant (environmental), in partial steady state or whether it can be neglected (partially or completely). Further simple model system to help guide understanding and use of the index analysis can be found in the S4 Supplementary Material.

### Input-response indices of the EGFR signalling pathway guide subsequent analysis of the key molecular species

To illustrate applicability and usefulness to large-scale systems, we next performed an index analysis of the epidermal growth factor receptor (EGFR) signalling cascade. The EGFR pathway is activated by binding of EGF to EGFR. Dimerised EGF-EGFR autophosphorylates and recruits a series of adaptor molecules called GAP, Grb2 and Sos. Signal transduction may occur via membrane-bound or internalised species; in addition it occurs via two major pathways: the Shc-dependent and the Shc-independent pathway. The two pathways, however, do not act independently from each other, since there are many molecules involved in both pathways. Both pathways eventually activate Ras-GDP, a well-known oncogene and the merging point of both pathways. Subsequent activation of Raf transduce the signal to the MAP kinase

cascade and finally to ERK. The output signal is double-phosphorylated ERK, which transiently increases as a response to the input stimulus. In view of the input (EGF) and the output (ERK-PP), the signalling cascade is sometimes termed the EGF–ERK-PP system. A graphical representation can be found in Fig I in S5 Supplementary Material.

While the principal transduction of the signal is well known, the relative importance of the different pathways (Shc-dependent and Shc-independent, membrane-bound vs. internalised forms) and its molecular constituents is still not well understood.

To index-analyse the EGFR signalling cascade, we chose a constant extra-cellular EGF stimulus of 50 nM and a time interval [0, 100] min, as in [20]. Fig 7A shows the temporal response of the ERK-PP output signal; the peak of the signal is reached within 3 min. The sensitivity-based indices were determined according to Eq (11). Fig 7B shows the sum of the ir-indices over time. We clearly identify three different phases: an initial peak (0–0.3min), a short high plateau (until 3min); followed by an extended and finally decaying low plateau. While the extended low plateau corresponds to the decay phase of the output, the initial peak and the short high plateau correspond to signal onset and steep increase.

Fig 8 shows all normalised ir-indices that exceed a threshold of $\delta_y$ = 10% at least once during the time span—20 out of a total of 112. The threshold value was chosen empirically, but also matched a gap in the decay of the maxima of the normalised input-response indices (see Fig 8D). In broad terms, Fig 8 shows an ordered appearance and disappearance of states, as might be expected. A closer examination reveals that at any point in time there exist often 3–5 states with a normalised ir-index above 10%, but always at least one index (see Fig II in S6 Supplementary Material). At any point in time, there are thus only a few dynamically important states responsible for the transduction of the input to the output. For further analysis, the 20 states with a maximal normalised ir-index exceeding 10% are coloured in light blue in the EGFR model cartoon in Fig 9. Interestingly, the 20 states all belong either to the membrane-bound and Shc-dependent receptor species (as opposed to the internalised or Shc-independent species), or the free cytosolic Ras, Raf, MEK and ERK species (as opposed to the internalised species).

Further observations can be gained from Fig 8: First, not all species that are absolutely necessary to transmit EGF to ERK-PP have a normalised ir-index exceeding 10%, including

**(A)** **(B)**

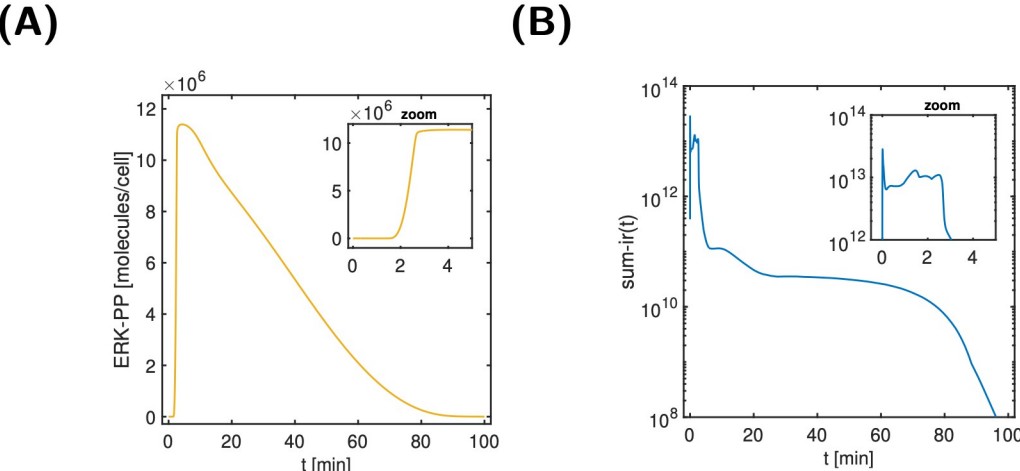

**Fig 7. Output signal (left) and sum of ir-indices evolving over time (right).** A: Transient increase of ERK-PP (output) in response to the EGF (input) stimulus. Inset zoom: signal activation occurs within 3 min. B: Sum of ir-indices over time, showing three phases: initial peak (0–0.3 min), high plateau (0.3–3 min), and low plateau including decay (3–100 min).

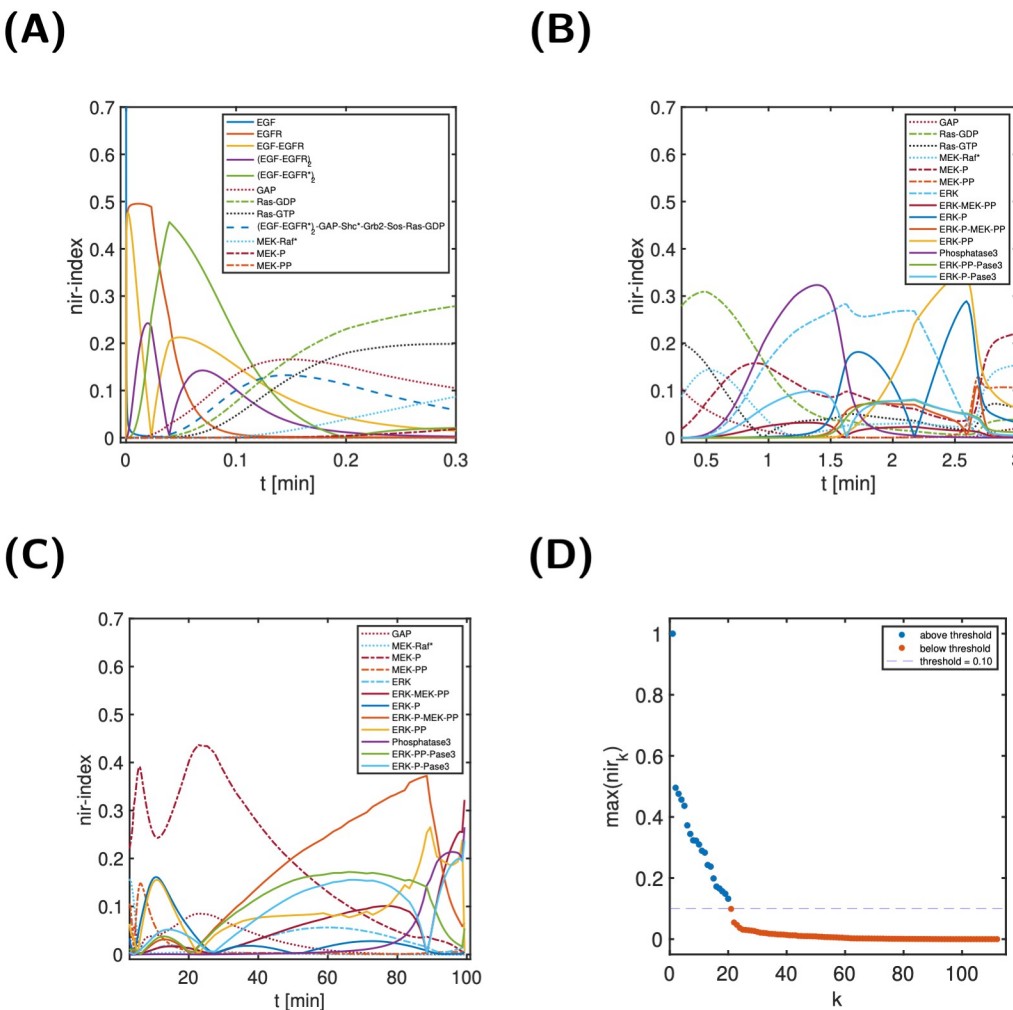

**Fig 8. Normalised ir-indices over time and ordered maximal value.** A, B and C: Normalised ir-indices for the three phases (0–0.3 min, 0.3–3 min, 3–100 min). Shown are all 20 indices with a maximal value exceeding $\delta_y$ = 10%. D: Order of decay of maximum value of the normalised ir-indices. All 20 states above the threshold of 10% are membrane-bound or free cytosolic forms and (as far as applicable) belong to the Shc-dependent pathway—as opposed to internalised forms and the Shc-independent pathway; see Table I in S6 Supplementary Material. The state just below the threshold is (EGF-EGFR*)$_2$-GAP-Shc*-Grb2-Sos.

adaptor proteins Shc, Grb2, Sos as well as Raf and MEK. Second, while Phosphatase3 is known to be involved in signal deactivation, its nir-index already peaks around 1.4 min and thus during signal activation. Phosphatase 1 and 2, however, do not seem to have a similar role during the activation phase. Finally, MEK-P and ERK-P-MEK-PP seem to be relevant during the deactivation phase (see Fig 8C), while we would rather associate them with the activation phase. The analyses below are guided by these observations. In addition, we computed the state classification indices and relative state approximation errors of all states not classified as dynamic, resulting in 2 environmental, 34 in partial steady state, 41 completely negligible, and 24 partially negligible states. Of note, this classification is not unique, with the largest overlap between completely negligible and in partial steady state/partially negligible. Proceeding as in Fig 1 (order: dyn, env, pss, cneg, pneg) resulted in a non-overlapping classification of 2 environmental, 34 in partial steady state, 20 completely negligible, and 2 partially negligible states.

## (A) membrane EGFR signalling    (B) internalised EGFR signalling

**Fig 9. Schematic with state classification of the signal transduction network focussing on the Shc dependent pathway for membrane EGFR signalling (Panel A) and internalised EGFR signalling (Panel B).** The schematics include the 20 state variables with largest maximum input-response index (light blue, see Fig 8D), environmental state variables (purple), state variables in quasi-steady state (green) and further state variables (dark blue). States being part of the membrane-bound and internalised pathway are coloured orange in panel (B). The red boxes mark the input and output state variables; coloured dots as part of reaction arrows indicate intermediate complexes. A similar graphic for the Shc-independent pathway is Fig I in S6 Supplementary Material.

In summary, the index analysis provide a classification of 80 of 112, or approximately 3/4 of the states.

### Phosphatases1–3 show very different behaviours and functions

Phosphatase3 (abbreviated P'ase3, when part of a complex) is involved in de-phosphorylation of ERK-P and ERK-PP (see also Fig 9). Fig 10A depicts the time course of key molecular species involved in the local Phosphatase3 network during signal activation. Once the kinase MEK-PP is present, it phosphorylates ERK via ERK-P to ERK-PP. Rather than observing an increase in ERK-P followed by ERK-PP, however, we first see a steep increase of ERK-P:P'ase3, indicating that Phosphatase3 immediately binds ERK-P to form a complex and subsequently de-phosphorylates it. Only when the level of Phosphatase3 decreases sufficiently, ERK-P levels increase markedly. As with ERK-P, double-phosphorylated ERK-PP is immediately bound by Phosphatase3 and subsequently de-phosphorylated, as can be inferred from the step increase of ERK-PP:P'ase3. Again, only when the Phosphatase3 level further decreases, the output signal ERK-PP increases to high levels. Thus, Phosphatase3 delays signal onset by sequestering ERK-P and ERK-PP; moreover, it controls the ERK-PP peak concentration as well as signal deactivation. This is confirmed by simulating the model with no Phosphatase3, as shown in Fig 10B. In summary, the described action of Phosphatase3 can be understood as a protection mechanism against activation of ERK-PP by spurious or random phosphorylation of ERK or ERK-P in the absence of a signal.

For Phosphatase1, Fig 11A show the normalised state classification indices. Since both, the env-index and the corresponding relative state error (see Fig IV in S6 Supplementary Material) are below the threshold, we classify Phosphatase1 as environmental. A further look at the

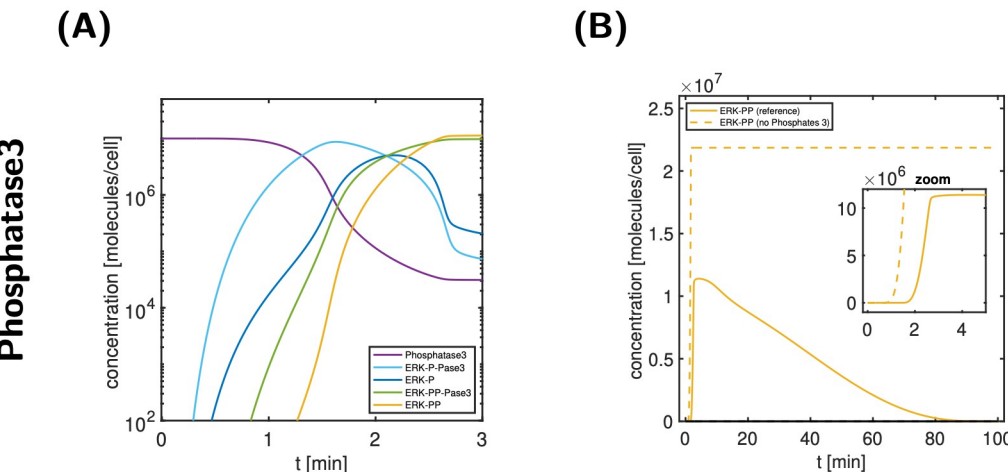

**Fig 10. Relevance of Phosphatase3.** A: concentration-time profiles of Phosphatase3 and relevant cytosolic ERK species during signal activation. B: comparison of the output ERK-PP for the reference simulation (solid lines) and a modified model (dashed lines) with no Phosphatase3. As a result, signal activation speeds up by roughly 1 min, in addition to lack of signal deactivation.

Phosphatase1 levels of the reference model (see Fig 11B) is confirmatory. In addition, the prediction of the modified model with constant Phosphatase1 levels from $t = 0$ on (i.e., as environmental state) are shown; the differences to the original model predictions are very minor.

We coloured all state variables classified as environmental in pink in the model scheme in Fig 9. We infer that only one other species—Prot, a coated pit protein that mediates receptor internalisation—is classified as environmental.

For Phosphatase2, none of the normalised state classification indices are below the threshold for all times; as shown in Fig 11C. A closer look at important species related to Phosphatase2 (see Fig V in S6 Supplementary Material) revealed that a noticeable fraction of Phosphatase2 is bound to internalised MEK, e.g., MEKi-P and MEKi-PP (10–12% for a substantial amount of time). This suggests that internalised MEK plays an important role in sequestering Phosphatase2. Fig 11D further supports this hypothesis: it shows the predictions of the reference model and a modified model with no internalised MEK species (all MEKi-species classified as cneg). Due to increased availability of free Phosphatase2, free MEK-PP levels are lower, and ERK-P bound MEK-PP levels are higher, the preceding species of ERK-PP. As a consequence, the signal is attenuated. This might indicate that the pool of *internalised* MEK plays a role in signal prolongation by sequestering Phosphatase2.

## Raf* dynamics is fast throughout activation and deactivation

The maximal value of the ir-index of Raf* is 0.17%≪10% (see Table I in S6 Supplementary Material), a surprisingly small number given the relevance of Raf* in signal transduction. Fig 12A and 12B show the normalised state classification indices and the corresponding relative state approximation errors for Raf*. Since both, the pss-index and the corresponding relative state error are below the threshold, we classified Raf* as partial-steady state. The B panel shows the prediction of the reference model and a modified model with Raf* in partial-steady state from $t = 0$; the profiles are indistinguishable. Thus, signal propagation through Raf* is 'instantaneous'.

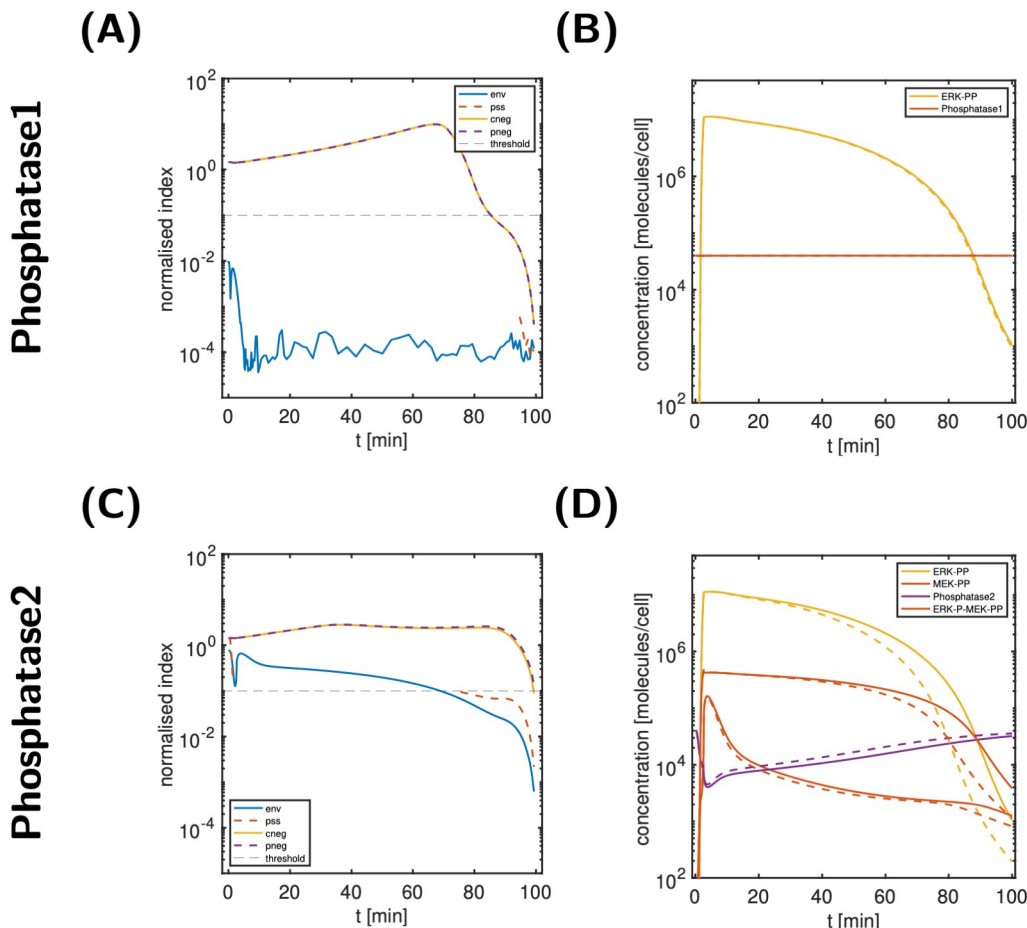

**Fig 11. Analyses of Phosphatase1 (top) & 2 (bottom).** A: State classification indices for Phosphatase1; B: comparison of the output ERK-PP and Phosphatase1 for the reference simulation (solid lines) and a modified model (dashed lines) with constant Phosphatase1 levels. C: State classification indices for Phosphatase2; D: comparison of the output ERK-PP, Phosphatase2 and two MEK-PP species for the reference simulation (solid lines) and a modified model (dashed lines) without MEK internalisation. Absence of internalised MEK increases free Phosphatase2 levels and thereby deactivates the signal more quickly.

We coloured all state variables classified as partial-steady state in light green in the model scheme in Fig 9. We infer that several other species, including a large fraction of internalised species, act on an instantaneous time scale.

## Ras-GTP* recycling is important for signal prolongation

The cycle of Ras-GDP activation and Ras-GTP inactivation is a central motif of the signalling cascade (see Fig 9). While Ras-GDP and -GTP have nir-indices above the threshold, the inactivated form Ras-GTP* has a nir-indices below the threshold for all times. Fig 12C shows the state classification indices for Ras-GTP*. Since none of them is below the threshold for all times, no further classification is possible (see also Fig 1). Fig 12D shows the prediction of the reference model and a modified model with Ras-GTP* partially neglected (pneg) from $t = 0$, i.e., removed from the network. The result is a response attenuation; the Ras-GDP profile clearly shows that the re-activation of Ras-GTP* to Ras-GDP does play an important role in

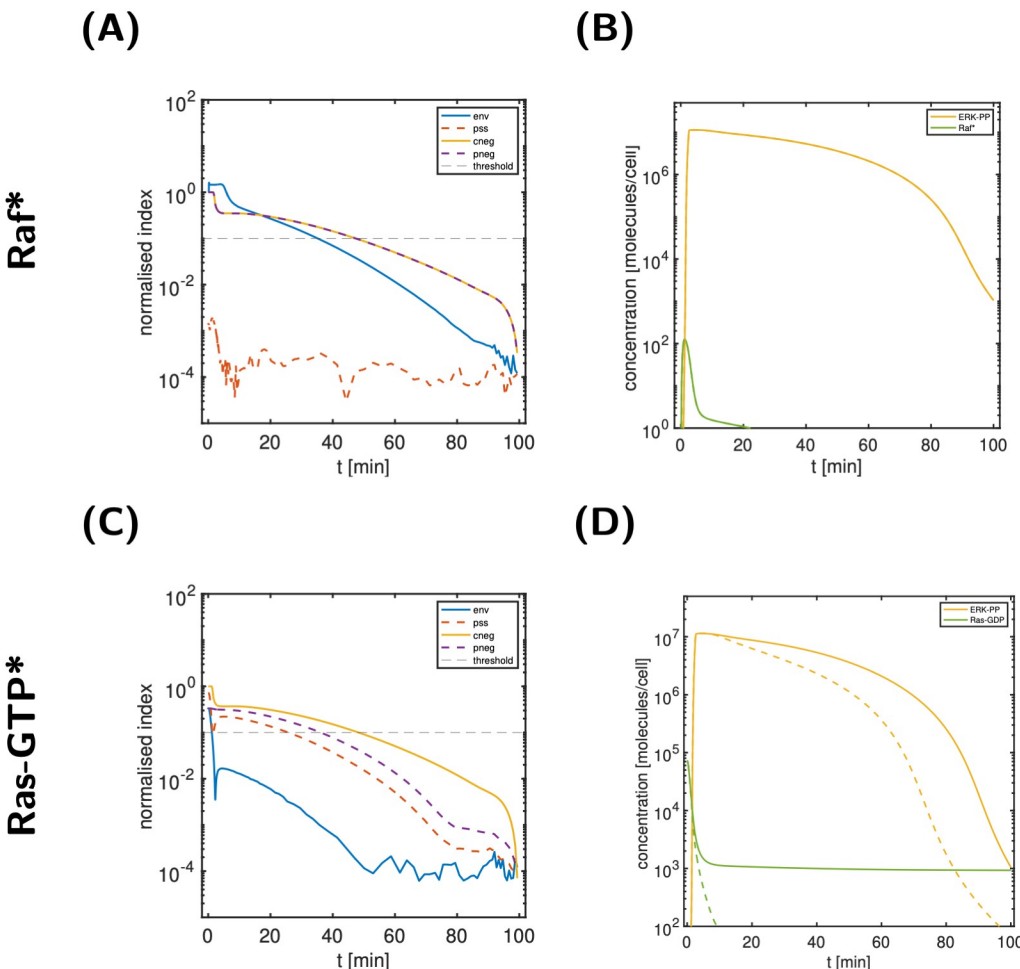

**Fig 12. Analyses of Raf\*, Ras-GTP\*.** A: State classification indices for Raf\*; B: comparison of the output ERK-PP and Raf\* for the reference simulation (solid lines) and a modified model (dashed lines, visually indistinguishable from the solid lines) with Raf\* in partial-steady state. C: State classification indices for Ras-GTP\*; D: comparison of the output ERK-PP and Ras-GTP\* for the reference simulation (solid lines) and a modified model (dashed lines) with partially neglected Ras-GTP\* (pneg).

signal prolongation by replenishing the pool of Ras-GDP molecules—starting already from 1 min onwards and remaining important even during signal decline.

## Degradation of internalised receptors-complexes results in faster decrease of Ras-GTP levels

Prot-mediated internalization and subsequent degradation of internalised receptor species plays an important role in output signal shut-down. Fig 9B shows six internalised EGFR species including degradation (out of 16 degradation reactions in total). For these six receptor species, degradation has a large impact on the output signal. Fig 13 shows the prediction of the reference model and a modified model with no degradation of these six species from $t = 0$ (realised by classifying the corresponding degradation products, e.g., (EGF- EGFR\*)$_2$- deg as completely negligible). Without degradation of these receptor species, the signal is strongly prolonged: An elevated concentration of internalised receptors increases the recycling rate to

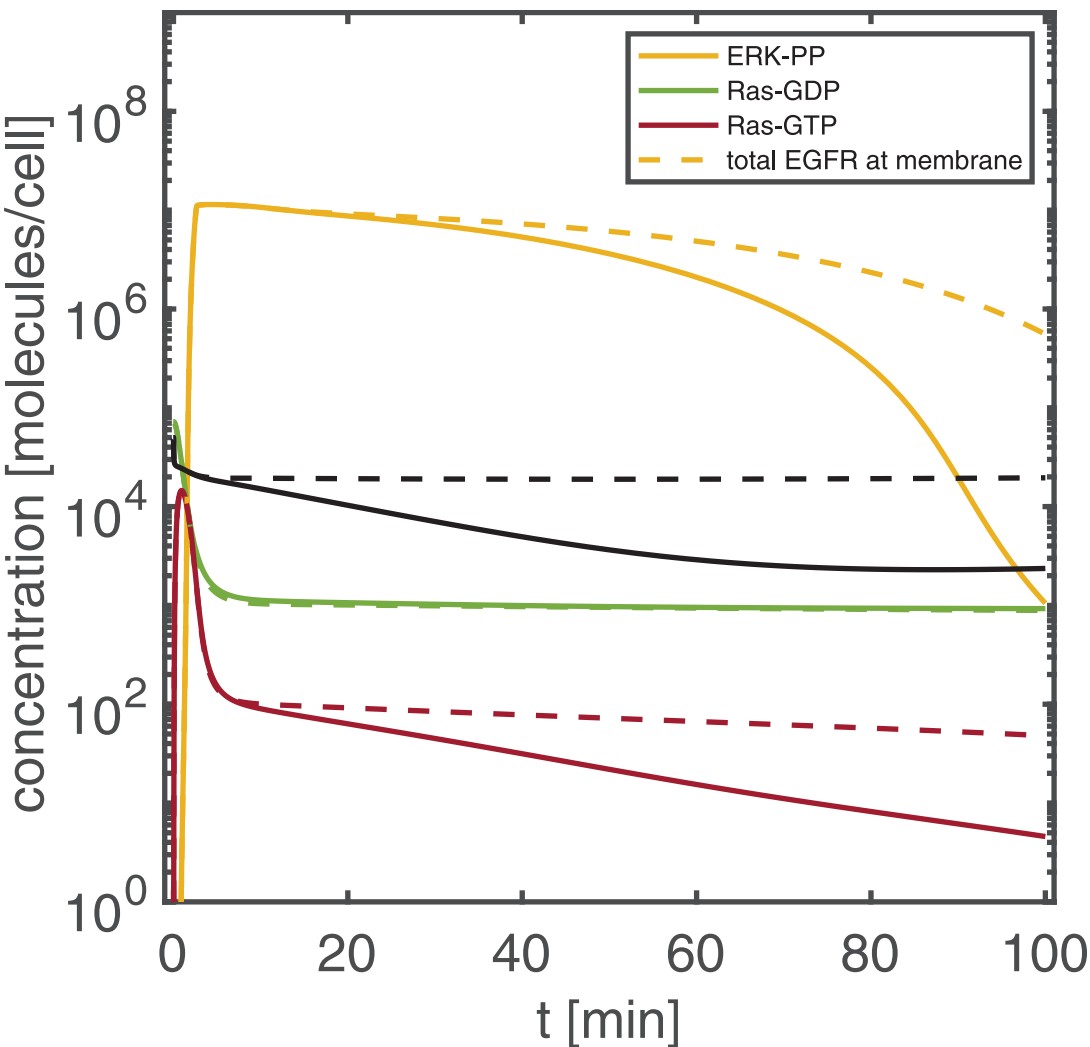

**Fig 13. Analyses of receptor degradation.** Comparison of the output ERK-PP and Raf* for the reference simulation (solid lines) and a modified model (dashed lines) with no degradation from six internalised receptor species (those with degradation reaction in Fig 9, realised by classifying the degradation products as cneg).

the membrane, impacting the total concentration of membrane-bound receptor species. While the cytosolic species Ras-GDP is hardly changed, elevated receptors levels eventually increase the activation rate from Ras-GDP to Ras-GTP via increased levels of (EGF-EGFR*)$_2$-GAP-Shc*-Grb2-Sos. Increased levels of activated Ras-GTP, finally, result in a prolonged output signal.

## Discussion

We defined the sensitivity-based input-response index and four state classification indices that together can be used to analyse signal transduction networks. All indices are (i) time- and state-dependent; defined (ii) for a specific input-output relationship; and (iii) for a specific magnitude of the input. We have illustrated the approach for simple small-scale models and

demonstrated for the EGFR signalling network, how index analysis can provide an in-depth view on the role and contribution of the different molecular species.

The sensitivity-based input-response index is defined in terms of two sensitivity coefficients. In sensitivity analysis, local sensitivity coefficients measure the impact of changes in parameters $p$ or initial conditions (also seen as input parameters) on a response variable $x_r$ as a function of time [4]. They are used, e.g., for parameter sensitivity analysis or to study the robustness of signal transduction pathways. A key challenge in sensitivity analysis is to account for scale-difference and to make them invariant to transformation of units and thus comparable. A common solution is to consider scaled sensitivity coefficients

$$G_{r,i}(t) = \frac{\partial x_r(t)}{\partial p_i} \cdot \frac{p_i}{x_r(t)}, \qquad (17)$$

resulting in a unitless quantity know as the logarithmic gain [26], or as control or response coefficient in metabolic control analysis [27, 28]. In signalling cascades, signals are often propagated through a series of activation steps. Activated forms temporarily rise from zero/very low concentrations to their maximum, while inactive forms almost vanish. In such a situation, the scaling above is highly problematic, since $x_r(t) \approx 0$ at initial times for active forms and $x_r(t) \approx 0$ at later times for inactive forms, resulting in ill-defined coefficients or numerical problems. Of note, metabolic control analysis has mainly and very successfully been applied to metabolic systems in steady state, where such problems are absent. The input-response index differs in two regards: (i) it considers the integral of the first factor in Eq (17) rather than the evaluation at a single point in time; and (ii) it scales with the controllability index of the $k$th state variable (see Eqs (11) and (13)), rather than the factor $p/x_r(t)$ in Eq (17). This makes a key difference and firmly links the input-response index to a specific input, ensures invariance under internal unit transformations; and at the same time resolves the above scaling issue.

In signalling cascades, some state variables are important for very short periods of time. Therefore if a state variable is dynamically important at some point in time (even, if only for a very brief period of time), we consider it to be not dynamically irrelevant, i.e., dynamically relevant. This motivates the choice of the maximum of the input-response index as the relevant characteristic. For applications to chronic progressive diseases, longer time periods are of interest. Then, slow changes in endogenous or exogenous factors may determine the rate of change in the system, while such changes are likely to be irrelevant on short time intervals. In this case, the metric might need to be changed, e.g., to the integral over the input-response index.

While the sensitivity-based input-response index is a measure of importance based on the original (unmodified) system, the state classification indices are measures involving a modification of the system. It is important to realise that a small state classification index solely indicates that the corresponding modification has a low impact on the output. This might be due to different reasons:

(i) the modification has a low impact on the concentration-time profile of the corresponding state—and as a consequence, also on the output;

(ii) the system is able to compensate for the impact of the modification on the state variable—and as a consequence, the impact on the output is low;

(iii) the state is not relevant for the input-output relationship—so even a larger impact on the state variable does not impact the output.

By using the state classification index in combination with the relative state approximation error, we aim to distinguish between (i) and (ii) for environmental and partial-steady state

classifications. The partially/completely neglected indices are defined to identify case (iii). It is, however, important to note that a low partially/completely neglected index may as well be indicative of case (ii). See the analysis and illustration of the parallel pathways model in the Section S4.1 in S4 Supplementary Material. This is also of relevance in EGFR signal transduction, where conflicting results have been reported concerning the importance of the Shc-dependent pathway [15, 29]. The authors in [29] state that the Shc-dependent pathway only plays an important role for low EGF concentrations. Their statement was based on a simple *in silico* knock-out study of the system. In contrast, the authors in [15] use sensitivity as well as flux analysis to conclude that signalling transduction proceeds primary through the Shc-dependent pathway. This contradiction was in part resolved in [30] by analysing the reactions of the EGFR signalling cascade model. They found that the production of Ras-GTP is indeed mostly mediated by the Shc-dependent pathway, while in case of the Shc knock-out, the Shc-independent pathway takes over the full activation of Ras-GTP. This analysis is easily reproduced with the completely neglected index (cneg): a knock-out of either all Shc-dependent or all Shc-independent species result in both cases in low relative errors on the output ERK-PP (5.5% and 1.1%, respectively).

In light of the low relative errors (and without knowledge on the pathway), one would conclude that neither the Shc-dependent nor the Shc-independent pathway is important. This conclusion, however, is falsified by the joint knock-out resulting in a 100% relative error on the output ERK-PP (no output at all). The example clearly illustrates that care has to be taken when interpreting the importance of states or entire pathways from (in-silico, but also in-vitro/vivo) knock-out studies.

Index analysis approaches the question differently. The input-response indices clearly highlight the important dynamic role of the Shc-dependent species. Importantly, the ir-indices do not require a modification (e.g., knock-out) of the system—in a certain sense, they measure the dynamic importance of states non-invasively. At the same time, our analysis shows that many states of the Shc-independent pathway are classified as being in partial-steady state. Hence, it is the "dynamic nature of the state variables" that is the most prominent difference between the Shc-dependent or Shc-independent pathway.

The precise impact of receptor internalisation and the role of internalised species is still up to debate [1, 15, 20]. We infer from Fig 9 that a prominent difference between the membrane-bound and internalised pathway is again the "dynamic nature of the state variables". No internalised species is classified as dynamic (large nir-index), while a large number is classified as partial-steady state. Using the partially/completely neglected index, we identified the relevance of receptor degradation from specific internalised receptor species (indicated by degradation reactions in Fig 9). Absence of receptor degradation from these species increases the pool of internalised receptors and thus receptor recycling to the membrane; this eventually results in higher levels of activated Ras-GTP and finally in signal prolongation. At the same time, we identified the role of internalised MEK in sequestering Phosphatase2 in complexes, lowering the deactivation rate of Raf* and thereby prolonging the signal.

A detailed study of the EGFR signalling pathway is presented in [20]. The authors quantified the relevance of *reactions* based on the concept of impact control coefficients, and the importance of proteins based on a fractional change of their *total* concentration. The time-dependent output was described by three characteristics: the amplitude, duration and integral of the ERK-PP profile. Index analysis allows a complementary view on the system. It focusses on individual state variables and on time, in contrast to reactions and lumped total concentrations. In [31] general principles that govern signal transduction are identified, with the central conclusion that collectively, kinases control amplitudes more than duration, whereas Phosphatases tend to control both. Our time-resolved index analysis of the Phosphatases of the EGRF

signalling network, in particular Phosphatase3, supports this conclusion and adds an additional detail: Phosphatases might also control the time to signal onset. In addition, we identified mechanistic principles underlying the conclusion including, e.g., the stoichiometric ratio of the Phosphatases and their binding partners (see paragraph on Phosphatases1–3 in the Results). While free Phosphatase1 levels are hardly impacted by complex formation with Raf*, free Phosphatase3 levels are reduced by three orders of magnitude; nearly 100% of Phosphatase3 is sequestered in complex with ERK-PP for a long time.

Overall, the input signal EGF exerts its impact on the output ERK-PP only during a very small time window (see Fig IIIA in S6 Supplementary Material); it does so by strongly impacting, i.e., controlling, state variables during this time frame. In combination with roughly seven orders of magnitude smaller observability indices (see Fig IIIB in S6 Supplementary Material), this can be interpreted as some robustness property of the signalling cascade. Random fluctuations of constituents in the absence of an input signal are unlikely to span several order of magnitude needed to spontaneously activate the cascade.

During the review process, some concerns were raised about artefacts in the EGFR signalling model [1] used in our analysis that do not reflect the most recent biological understanding. Within the model, the EGFR homo-dimer bings to either one SHC or one GRB2 molecule. It has been shown, that each EGFR protomer in the dimer has multiple phosphotyrosine residues that could bind SHC1 and multiple residues that could bind GRB2, and on the EGFR homo-dimer, there is no limit to only one at a time across both individual EGFR proteins [32, 33]. Furthermore, the biological correspondence of the RAS-GTP* and RAF* states was questioned.

The numerical effort to compute all indices for the illustrative model systems is in the seconds, and for the EGFR signalling cascade a few hours (on a decent laptop). For the ir-index, the most demanding part is the solution of the sensitivity equation to determine the controllability index (a single simulation on $[t_0, T]$) and the observability index (for each $t^*$ a simulation on $[t^*, T]$); see Eq (13) and S1 & S2 Supplementary Materials. We chose the number of $t^*$ values to be identical to the (adaptively choosen) timepoints of the numerical scheme when integrating the reference solution. This guaranteed good approximation quality of the integral over the observability index (see Eq (13)) by the trapezoidal rule—in addition to smooth visual plots. Since the Jacobian of the right hand side of reaction kinetic models is typically sparse, we exploited this advantageous feature numerically (by using a sparse representation of the Jacobian matrix). For the state classification indices, the numerical effort is in simulating the modified system of ODEs (repeatedly for each $t^*$ a simulation on $[t^*, T]$). Of note: in particular for the partial steady state index, the numerical integrator can face problems in cases, when a partial steady state assumption will result in a blow-up or when determining consistent initial conditions for the DAE system. In the first case, we simply assigned an NaN value to the corresponding index (the state would not be classified as pss anyway), while in the second case, we retried with a different initial guess of the consistent initial condition (up to three times). This approach was successfully used in all considered examples.

If in case of very large ODE systems, e.g., when automatically generated by rules as impressively illustrated in [13], the solution of the sensitivity equation is challenging, the empirical input-response indices [19] might provide an alternative.

Index analysis naturally links to model reduction. In [19], we introduced an empirical input-response index by building and expanding on the concepts of controllability and observability from control theory. Based on the empirical index, we proposed an iterative model reduction scheme. In application to the blood coagulation network, we illustrated its usefulness in a clinically relevant setting. A key feature of the proposed model reduction technique is its reference to a local regime in the state space (by defining a reference input in addition to

input/output state variables). We demonstrated the advantage of such a local approach by identifying different reduced models based on different reference inputs. For the given application, this allowed to understand the lack of impact of certain genetic modifications for the outcome of the standard blood coagulation test and the presence of impact for a modified test. In the present work, the focus is rather on understanding a given signal transduction network. At the same time, the index analysis provides new insights and opens new possibilities for future model reduction approaches. By introducing state classification indices jointly with corresponding relative state approximation errors, we clearly discriminate between the impact of an approximation on the output and on the state itself. The newly introduced state classification indices provide further options for the model reduction approach proposed in [19]. In addition, we have extended the concept of input-response indices from a fixed reference model to an ensemble of models (based on parameter variability) representing, e.g., inter-individual or inter-cellular variability [34]. This paves the way for applications in the drug discovery and development process. In [34], we demonstrate how to use the sensitivity-based input-response index in combination with a model reduction approach detailed in [19] to reduce a large-scale system biology model of the blood coagulation cascade. The result is a small-scale pharmacokinetik/pharmacodynamic model (a Warfarin-INR model) suitable for the analysis of clinical data. Application to clinical data in a therapeutic drug monitoring context is work in progress.

All in all, we believe that the proposed index analysis approach substantially broadens our means to analyse and understand complex signal transduction models in systems biology.

## Supporting information

**S1 Supplementary Material. Numerical computation of ir- and state classification indices.** (PDF)

**S2 Supplementary Material. Pseudocode of numerical computation of ir- and state classification indices.** (PDF)

**S3 Supplementary Material. How to setup a new model for Index Analysis.** (PDF)

**S4 Supplementary Material. Index analysis for additional small-scale illustrative model systems.** Section S4.1. Index analysis for parallel pathway model. Section S4.2. Index analysis for enzyme substrate model. (PDF)

**S5 Supplementary Material. EGFR systems. Fig I. Simplified illustration of EGFR signalling cascade. Fig II. EGFR signalling network prior to a stimulus.** An illustration of state variables that are not in steady state prior to any EGF stimulus. For details on the model, see Material Section in the main article. (PDF)

**S6 Supplementary Material. Additional figure and tables for the index analysis of the EGFR system. Fig I. Schematic with state classification of the signal transduction network focussing on the Shc-independent pathway**, including the state variables with large maximum input-response index (light blue, see Fig 4D), environmental state variables (purple), state variables in partial-steady state (green) and further state variables (dark blue). States being part of the membrane-bound and internalised pathway are coloured orange in panel (B). The red boxes mark the input and output state variables. Note that the difference to the

Shc-dependent pathway is the absence of Shc (adaptor protein between GAP and Grb2). **Fig II. Number of normalised ir-indices above the threshold of 10% as a function of time. Fig III. Sum of contr- and obs-indices evolving over time.** Left: Sum of contr-indices over time, showing three phases: initial sharp peak (0–0.3 min), second prolonged peak (0.3–3 min), and a slow, still incomplete recovery period (3–100 min). Right: Sum of obs-indices over time, showing three phases: initial sharp decline (0–1.5 min), marginal increase (1.5–20 min), and a strong decline (20–100 min). Note the very different scales on the y-axis. **Fig IV. Relative state approximation errors for Phosphatase1. Fig V. Time course of important species related to Phosphatase2. Table I. Normalised input-response indices for the EGFR system; sorted according to their maximum.**
(PDF)

## Author Contributions

**Conceptualization:** Wilhelm Huisinga.

**Data curation:** Jane Knöchel.

**Formal analysis:** Jane Knöchel.

**Software:** Jane Knöchel, Wilhelm Huisinga.

**Supervision:** Charlotte Kloft, Wilhelm Huisinga.

**Writing – original draft:** Jane Knöchel, Wilhelm Huisinga.

**Writing – review & editing:** Jane Knöchel, Charlotte Kloft, Wilhelm Huisinga.

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
