## [Decision Letter · Decision Letter 0]

26 Jun 2023

Dear Prof. Dr. Huisinga,

Thank you very much for submitting your manuscript "Index analysis: an approach to understand signal transduction with application to the EGFR signalling pathway" for consideration at PLOS Computational Biology.

As with all papers reviewed by the journal, your manuscript was reviewed by members of the editorial board and by several independent reviewers. In light of the reviews (below this email), we would like to invite the resubmission of a significantly-revised version that takes into account the reviewers' comments.

We cannot make any decision about publication until we have seen the revised manuscript and your response to the reviewers' comments. Your revised manuscript is also likely to be sent to reviewers for further evaluation.

Sincerely,

James R. Faeder

Academic Editor

PLOS Computational Biology

Jason Haugh

Section Editor

PLOS Computational Biology

Reviewer's Responses to Questions

**Comments to the Authors:**

Reviewer #1: In this work, Knoechel et al. introduced a new method, called ‘index analysis’, to study how the output of a kinetic model is affected by different types of perturbations on the state variables (model species). The authors also demonstrated the applications of this method with a simple toy model and a very complicated EGFR signaling model. The ‘index anlysis’ method would be interesting for the field of computational systems biology. It is also potentially useful for the study of dynamical biological systems. However, the presentation of this work needs further improvement. It is not very compelling how much insights can one learn from this method in practice. A comparison to existing sensitivity analysis methods is missing, which makes one difficult to judge the value of this method.

Major comments:

1. It is not clear for me what are the advantages and disadvantages of the ‘index analysis’ compared to the existing sensitivity analysis methods. Time-dependent sensitivity analysis can be performed with control coefficients and other sensitivity analysis methods as well. As the authors have not done any comparison with previous methods. I’m not sure whether the ‘index analysis’ proposed in this study could reveal some insights that cannot be seen by published time-dependent sensitivity analysis methods?

2. The introduction to the different indices is not very straight-forward. It is not easy for the readers to follow different definition of state classification indices based on their names. Maybe the authors can invent some intuitive and easy-understandable terms. For example, equation 15 already define the index. The so-called 4 state classification indices correspond to different perturbations on a selected state variable: setting the variable with a constant value (envk); describing the dynamics of the variable with an algebraic equation (pssk); removing the reactions that contains the variable as a reactant (pnegk); setting the state variable with a value 0 (cneg, correspond to knockout). I don’t understand what are the differences between pnegk and cneg? In most cases, I would expect the same results for pnegk and cneg. Is pnegk really needed? Here I would suggest the authors to simplify different types of state indices, for example, focus on only two perturbations: setting the select variable with a value of zero (cneg, KO) and non-zero value (envk). It seems more indices also causes confusions.

3. Although there are elaborated descriptions about the analysis of EGFR signaling, it is not clear for me that the state classification indices analysis results contributed to the understanding of the key molecule species as the authors stated. For example, In the paragraph line 336-351, the authors discussed the role of phosphatases 3, but what is the relevance of ‘index analysis’ results in this context? Isn’t it already clear based on the dynamics with the reference state?

4. The authors set an important threshold value 0.1 for state classification based on indices. Without showing many examples, it is not clear how this value can be generally applied in other cases? How this value is determined?

Minor comments:

1. It’s not clear for me how to perform “partial steady state index (pssk)” analysis in practice. The authors suggest to removing the ODE for the kth variable (kth ODE =0) and modify the ODE systems with DAE system. How this is implemented in practice. It will be better to illustrate it with the toy example by explicitly writing the corresponding equations of the ODE/DAE systems for different state classification indicies.

2. The state classification indicies results are highly dependent on different parameter values and the signal input. In addition, the ODE model parameters are not always identifiable. Therefore, one should be cautious to link the sensitivity analysis to the model reduction application. This could be explained in the discussion part.

3. There are some typos in the manuscript. For example,

Page 12, line 183 ‘partial-staedy’ -> ‘partial-steady’

Page 41, Fig. 2B: ‘kof’ -> ‘koff’

Fig. 12B, same color are used for ERK-PP and Raf*

Reviewer #2: Within this manuscript, the authors extend upon their previous work adapting concepts from control theory to the characterization of biochemical networks. Additionally, they apply it to signaling networks, using a well-known EGFR pathway model as a test case.

The work appears to be very rigorous technically. (This reviewer has not taken the time to validate their derivations or their computational work. If requested by the editors, this reviewer would be happy to jump in for a deeper technical review.). The authors are, at minimum, familiar with the relevant literature, grasp the key concepts very well, and communicate everything at a high level.

This reviewer has a few concerns.

Much of the manuscript either deals with illustrating the approach with a very simple model for illustrative purposes (Figures 2, 3, 4, 5, 6) or with illustrating the approach on a well-known, more complex model (Figures 7, 8, 9, 10, 11, 12, 13). Although walking through an approach can be helpful and informative, it can read more like a tutorial or a demonstration than a research paper. i.e. it is difficult to be excited about results from arbitrary models or flawed models; the value comes from learning about the method that could then be applied to one’s system of interest. The illustrative examples are fine, but they are not necessarily convincing that this is an improvement over other methods that are available (and that the authors discuss in the introduction).

One challenge I have with the paper is not the authors’ fault, but is that the well-known EGFR model that is used as a test case has many artifacts that are not realistic. Somewhat concerning, these artificial constructs play a big role in the illustrative walk through. For example, using a computational tool to determine which of two processes is more important in a complicated dynamical system is a reasonable question to pursue. However, the discussion of SHC-dependent and SHC-independent pathways and its inference from the model is somewhat awkward to read because the model itself is not well-suited to study the biological problem. (Within the model, the EGFR homo-dimer binds to either one SHC molecule or one GRB2 molecule. In the real world, each EGFR protomer in the dimer has multiple phospho-tyrosine residues that could bind SHC1 and multiple residues that could bind GRB2, and on an EGFR homodimer, there is no limit to only one at a time across both individual EGFR proteins.) Thus, the tool may be useful for studying a model in terms of which reactions contribute to a given model output, but the example focuses on a biological question that cannot be answered with this model because the model is poorly designed to study that problem.

Similarly, the RAS-GTP* and RAF* states of this model also seem to be artificial constructs. They do not have a clear biological equivalent. It is not clear why they were introduced (my guess is the original model included them to obtain the desired output). So discussions about RAS-GTP* and RAF* again discuss artificial aspects of the model.

My biggest concern is it does not seem that the manuscript makes a strong case that these indices are needed or offer any benefit to other techniques. The authors conclude their abstract by stating that they envision some beneficial applications (comparing healthy and diseased, model reduction, changing how models are used in drug discovery/development), but without more demonstration of the power of this method, that speculation reads very aggressive.

Suggestions for strengthening the manuscript:

One straightforward way to improve the manuscript would include some type of comparison of how these different reactions are ranked as important by these indices compared to how they would be ranked by the other methods the authors discuss in lines 36-66. Direct comparisons may not be possible – but a table or schematic that compares how the different approaches discussed would characterize the EGFR and/or simple model, and the relative benefits and/or costs of each could be useful.

Most powerful would be if there is a case where the index analysis provides a biological insight that can be validated (even with existing data). That may be hard with the EGFR model (due to its artifacts) – but with the extensive EGFR kinetic signaling experimental literature, are there any things from this analysis that can be compared to existing data? Does the model help clarify anything experimentally? Or is it only useful at figuring out what is going on within the model (which here is unfortunately full of artifacts).

Also helpful – if the analysis suggests new experiments, what are they? If the model is useful, stating what experiments would test model-based insights would be helpful for evaluating if the index analysis is creating new insights. Stating suggested experimetns is the type of next-step that is commonly expected in manuscripts like this. (My apologies if I missed it, I only found the string "experiment" in the text once.)

Additional suggested edits:

Line 216 lists “Table ??”. I think it means to say Fig 2B.

Fig 8A and Fig 8B. Both “GAP” and “RAS-GTP” seem t ohave the same color/dash line type, so they look indistinguishable in my PDF. They should be indicated with a unique color/line type combination.

Fig 12B: it states in the legend that there should be two reference trajectories in solid lines and two modified model trajectories in dashed lines, but I only see two solid lines in Fig 12B.

Line 391: states Figure 12A, but I believe it means to say Figure 12C.

Line 393: states Figure 12B, but I believe it means to say Figure 12D.

Reviewer #3: Knöchel and colleagues introduce a framework that introduces several indices for the analysis of mathematical models of biological systems. They demonstrate the utility of these indices in terms of model simplification and interpretation on a small cascade example as well as a previously published, more complex model of EGFR signaling. Both aspects are relevant and timely and I am convinced that this work is an important contribution to the field. Accordingly, I would like to express my support to publish this after some minor revision

## Comments

1. Would be nice to describe the implementation of how these coefficients are computed numerically in a bit more detail

2. The introduction has a rather strong focus on model simplification, but there are a couple of related approaches for model interpretation that might be worth mentioning (could also be something for the discussion):

- https://doi.org/10.1101/2023.05.02.538686

- https://doi.org/10.1038/msb.2008.74

- elasticity and other coefficients, see e.g., https://doi.org/10.1049/iet-syb.2011.0015

- https://doi.org/10.1073/pnas.1414026112

- https://doi.org/10.15252/msb.202210988

- https://doi.org/10.1101/2021.01.26.428266

## Minor Comments

- l8 "indices" is rather unspecific, maybe give them a better name?

- eq (4) a bit surprising that u is the same dimension as x and the solution is additive. sounds like a rather specific stimulus/perturbation, so it might be worthwhile using a different symbols

- my impression is that y_mod (eq 15) and x_mod (eq 16) are not sufficiently well introduced

- l222 table ref broken

- l250 In contrast, give_n_ ...

**Have the authors made all data and (if applicable) computational code underlying the findings in their manuscript fully available?**

Reviewer #1: Yes

Reviewer #2: Yes

Reviewer #3: Yes

PLOS authors have the option to publish the peer review history of their article (what does this mean?). If published, this will include your full peer review and any attached files.

Reviewer #1: No

Reviewer #2: No

Reviewer #3: No
---

## [Decision Letter · Decision Letter 1]

27 Nov 2023

Dear Prof. Dr. Huisinga,

Thank you very much for submitting your manuscript "Index analysis: an approach to understand signal transduction with application to the EGFR signalling pathway" for consideration at PLOS Computational Biology. As with all papers reviewed by the journal, your manuscript was reviewed by members of the editorial board and by several independent reviewers. The reviewers appreciated the attention to an important topic and agreed that the method introduced in this paper may potentially be useful to the systems biology community. There was a split decision among the reviewers, however, with one reviewer feeling that the computational model of the EGFR system, which is the focus of an extended case study using the presented method, is sufficiently flawed that any conclusions from the analysis are not likely to be biologically meaningful. The editors feel that, given that the primary contribution of this paper is to show how the method could be used to analyze models of signal transduction in general, their choice to apply the method to a well-known and reproducible model of EGFR signaling is sufficient for the present purpose and we are likely to accept this manuscript for publication. Given the reviewer's concerns, we would appreciate if the authors could, as they have offered to do, add a short paragraph to the Discussion section of the paper that addresses some potential biological limitations of the EGFR model they have used in their case study based on the suggestions of Reviewer 2.

Sincerely,

James R. Faeder

Academic Editor

PLOS Computational Biology

Jason Haugh

Section Editor

PLOS Computational Biology

Reviewer's Responses to Questions

**Comments to the Authors:**

Reviewer #1: I'm satisfied with the authors' revision.

Reviewer #2: The point of peer review is for the authors of a manuscript to get comments from their scientific peers. The peers say what is good & what is bad, and the authors are supposed to work to make improvements to the manuscript that address these concerns. When it works, peer review makes the paper better. Reviewing papers is a lot of work, it isn't particularly fun to do, and the only reason to do it as a reviewer is to help make the field better.

The paper was initially rejected with request for significant revisions. The revised paper does not seem to have significant revisions. Several important issues raised by the reviewers are dismissed, and/or addresed superficially. In one case, after the authors were directed to several problematic areas of the model they are studying and choose to focus their efforts upon, the authors do not address these issues but rather ask for references (which are easy to find, as this is 100% standard biology that was butchered in the model). Thus, the authors do not seem to understand the biology they claim their methods are here providing "answers" to questions in the field.

The revision appears to have primarily involved edits to improve clarity, to have some work done to introduction and discussion, but not to have more work that adds to solidify the body of work as in the manuscript.

One of my major concerns is that a significant part of the analysis digs into a old and flawed EGFR signaling model. The previous review provided a few clear examples of where it is flawed, including there being multiple GRB2 and SHC1 binding sites, RASGTP* and RAF* being artifacts of the model, etc. The reviewers do not address these, but rather state that others have used this model and a (failing) company has put work into this model, seeming to imply it must be good.

Particularly concerning is that much of the demonstrations focus on studying the artifacts of the model and claiming it is useful to EGFR biology.

The authors respond with an argument to the effect of "others have done it, so why can't we?" and by pointing out that the biotech Merrimack has used this model. That others do something, of course, does not mean that it's a good course of action.

It also doesn't take much time on google to find that Merrimack is struggling. (This is the biotech mentioned in response to review that claims this model must be good because it is used by a drug company). It also doesn't take much work to find evidence in the literature for the things mentioned about GRB2 and SHC1 promiscuity... for example,

https://www.nature.com/articles/nature12308

https://www.nature.com/articles/nature04177

and any current review on RAS activation should show that RASGTP and RASGTP* (and RAF*) are artificial constructs.

The authors' inability to find experimental biology papers that are relevant to the claims of the manuscript is highly concerning. The authors continued claim that unanswered questions in EGFR biology can be answered by studying this (flawed) model with their method is also undermined by their inability to show that they understand actual EGFR biology.

Next time, the authors should spend more time pursuing the recommendations of peer review. Otherwise, just post to bioRxiv and let the work just be out there without going through peer review.

Reviewer #3: The authors have adequately addressed all my concerns.

**Have the authors made all data and (if applicable) computational code underlying the findings in their manuscript fully available?**

Reviewer #1: Yes

Reviewer #2: Yes

Reviewer #3: Yes

PLOS authors have the option to publish the peer review history of their article (what does this mean?). If published, this will include your full peer review and any attached files.

Reviewer #1: **Yes: **Zhike Zi

Reviewer #2: No

Reviewer #3: No

Figure Files:

Data Requirements:

Reproducibility:

References:

---

## [Editor Report · Decision Letter 2]

21 Dec 2023

Dear Prof. Dr. Huisinga,

We are pleased to inform you that your manuscript 'Index analysis: an approach to understand signal transduction with application to the EGFR signalling pathway' has been provisionally accepted for publication in PLOS Computational Biology. We appreciate the statement you have added to address the reviewer's concerns and hope that they may lead to further discussion among those working in the field.

Best regards,

James R. Faeder

Academic Editor

PLOS Computational Biology

Jason Haugh

Section Editor

PLOS Computational Biology

---

## [Editor Report · Acceptance letter]

14 Jan 2024

PCOMPBIOL-D-23-00183R2 

Index analysis: an approach to understand signal transduction with application to the EGFR signalling pathway

Dear Dr Huisinga,

I am pleased to inform you that your manuscript has been formally accepted for publication in PLOS Computational Biology. Your manuscript is now with our production department and you will be notified of the publication date in due course.

With kind regards,

Zsofia Freund
